# Identification of Potential Proteinaceous Ligands of GI.1 Norovirus in Pacific Oyster Tissues

**DOI:** 10.3390/v15030631

**Published:** 2023-02-25

**Authors:** Chenang Lyu, Jingwen Li, Zhentao Shi, Ran An, Yanfei Wang, Guangda Luo, Dapeng Wang

**Affiliations:** Department of Food Science and Technology, School of Agriculture and Biology, Shanghai Jiao Tong University, Shanghai 200240, China

**Keywords:** human norovirus, *Crassostrea gigas*, specific binding ligands, tumor necrosis factor, intraflagellar transport protein

## Abstract

Human norovirus (HuNoV) is the leading foodborne pathogen causing nonbacterial gastroenteritis worldwide. The oyster is an important vehicle for HuNoV transmission, especially the GI.1 HuNoV. In our previous study, oyster heat shock protein 70 (oHSP 70) was identified as the first proteinaceous ligand of GII.4 HuNoV in Pacific oysters besides the commonly accepted carbohydrate ligands, a histo-blood group antigens (HBGAs)-like substance. However the mismatch of the distribution pattern between discovered ligands and GI.1 HuNoV suggests that other ligands may exist. In our study, proteinaceous ligands for the specific binding of GI.1 HuNoV were mined from oyster tissues using a bacterial cell surface display system. Fifty-five candidate ligands were identified and selected through mass spectrometry identification and bioinformatics analysis. Among them, the oyster tumor necrosis factor (oTNF) and oyster intraflagellar transport protein (oIFT) showed strong binding abilities with the P protein of GI.1 HuNoV. In addition, the highest mRNA level of these two proteins was found in the digestive glands, which is consistent with GI.1 HuNoV distribution. Overall the findings suggested that oTNF and oIFT may play important roles in the bioaccumulation of GI.1 HuNoV.

## 1. Introduction

Human norovirus (HuNoV) is the major foodborne pathogen causing nonbacterial acute gastroenteritis in all ages worldwide [1,2]. HuNoV is transmitted primarily through human-to-human contact or by consuming contaminated water or food [3]. Approximately 14% of HuNoV outbreaks can be attributed to foodborne transmission [3]. Common food carriers include fresh or frozen soft fruits and vegetables (strawberries or lettuce), ready-to-eat foods (sandwiches or salads), and uncooked or raw seafood such as cockles, mussels, clams, scallops, and oysters [4,5,6,7]. The oyster is a filter feeder that enriches HuNoV particles from contaminated water into the oyster tissues. However, conventional clean seawater purification cannot effectively reduce HuNoV in oysters [8,9], and this coupled with the fact that oysters are often eaten raw makes them an important vector for HuNoV transmission.

Understanding the pattern and mechanism of HuNoV bioaccumulation in oysters can help control and reduce unwanted epidemics caused by consuming contaminated oysters. The search for HuNoV receptors and ligands has been greatly hampered by the lack of a stable and already imitated in vitro cultivation system for HuNoV [10,11]. However, several studies have demonstrated the presence of specific binding ligands for HuNoV in oyster tissues [12,13,14,15,16,17,18,19]. The binding of HuNoV to oyster tissue is commonly thought to be associated with specific HBGA-mediated interactions [12,13,14,15,17]. Other potential candidates involved in the accumulation of HuNoV in oysters have also been proposed, such as carbohydrate structures with a terminal N-acetylgalactosamine residue [12], sialic acid-like structures [16], and the newly discovered proteinaceous ligands, oHSP 70 [19].

On the one hand, there were multiple specific binding ligands in oyster tissues, and their distribution patterns differed in different tissues [12,16,19]. On the other hand, one specific binding ligand could bind multiple genotypes of HuNoV, but its binding ability differed [17,20,21]. The reasons mentioned above may lead to the different bioaccumulation efficiency and purification capacity of oysters for different genotypes of viruses. Moreover those mechanisms may also cause the different bioaccumulation patterns of different genotypes of HuNoV in different oyster tissues.

In our previous study, a ligand mining platform for norovirus using a bacterial cell surface display system (BSDS) was constructed [18,19]. As shown in Figure 1, the system uses *Escherichia coli* (*E. coli*) as a cellular vector, the N-terminal end of icosahedral nucleoprotein as an anchoring protein, and the P protein of HuNoV as a passenger protein, with a thrombin-recognizable fragment linked between the two to construct a pseudovirus with a large number of displayed P proteins on the *E. coli* cell surface [22,23]. The P protein in this system retains a similar ability to bind to the receptor as the viral particle [22] and can be used as a bait protein to target the prey protein (the specific binding ligands) out of the complex matrix using the principle of pull down. With the help of BSDS, an oligosaccharide ligand (H2N2F2) was identified from lettuce by glycomics [18], and a proteinaceous ligand (oyster heat shock protein 70, oHSP 70) was mined from oyster tissues by proteomics in our previous studies [19].

Although GII.4 HuNoV was the epidemic strain over past decades, GI HuNoV was more often implicated in shellfish-related outbreaks [12]. The GI.1 HuNoV is more likely to be enriched in oyster tissues than GII.4 HuNoV and also had a longer survival time [20,24]. In addition, the bioaccumulation pattern of GI.1 HuNoV did not strictly follow the distribution pattern of oHSP 70 or HBGAs in oyster tissues. Therefore we hypothesized that there are more specific ligands for GI.1 HuNoV in oyster tissues.

This study constructed a proteinaceous ligand library of oyster tissues using BSDS of GI.1 HuNoV P protein. The oyster tumor necrosis factor (oTNF) and oyster intraflagellar transport protein 74-like protein (oIFT) were selected for ligand evaluation according to a subcellular localization prediction. The interaction between the two recombinant ligands and the P protein of HuNoV was evaluated by ELISA. The mRNA of these two candidates in different tissues was detected by RT-qPCR. Our study built a proteinaceous ligands library for GI.1 HuNoV in oyster tissues, identified potential ligands, and provided new insights into the bioaccumulation mechanism of foodborne viruses in the Pacific oyster tissues.

## 2. Materials and Methods

### 2.1. Oyster Source and Acclimation

From December 2020 to September 2021, Pacific oysters were purchased monthly from an online retail shop (BEISILING, Rushan, Shandong, China). All the oysters (weighing between 90 g and 120 g) had Aquaculture Stewardship Council (ASC) labels and were traceable from the farm to our lab. The oysters were harvested from the farm (Rushan, Shandong, China) the same day we ordered them and transported in temperatures between 0 °C to 6 °C. A 50 L laboratory-scale experimental depuration system equipped with biological filters and a recirculating machine was used. Instant ocean salt (Yi’er BE Co., Ltd., Guangzhou, China) was added to pure water to a final concentration of 1.8% (m/V) to prepare the artificial seawater [25]. After washing, oysters were placed and kept in artificial seawater at room temperature for 24 h, as described earlier [26,27]. Oysters with better vigor were selected, and tissues were taken out respectively, including the mantle, heart, gills, and digestive glands. All tissues were stored at −80 °C for further use.

### 2.2. Protein Extraction

The mantle, heart, gills, and digestive glands (200 mg) were homogenized 15–30 times with a pre-cooled tissue grinder (Scientz, Ningbo, China), respectively. Protein was extracted by a Total Protein Extraction Kit (Sangon Biotech Co., Ltd., Shanghai, China) according to the protocol. The collected protein samples were stored at −80 °C for further use.

### 2.3. Construction of BSDS

The BSDS was used to capture the potential ligands of GI.1 HuNoV (Figure 1). The P protein of GI.1 HuNoV was anchored on the surface of *E. coli* BL21(DE3) with the N terminal (InaQn) of the ice nucleoprotein of *Pseudomonas syringae*. At the same time, an amino acid sequence that could be recognized by thrombin (TB) was added between the anchor protein and the P protein of GI.1 HuNoV. Therefore, the P protein could be separated from the surface of the host bacteria by thrombin digestion, to pull down the protein ligands of GI.1 HuNoV from the oyster tissue. *E. coli* BL21(DE3) with pET28a-inaQn-TB-I.1p (P) was constructed as described earlier [23]. The *E. coli* BL21(DE3) with pET28a-inaQn-TB-II.4p was used as the positive control, and the *E. coli* BL21(DE3) with pET28a-inaQn-TB (T) was constructed as the negative control. The recombinant *E. coli* BL21(DE3) was cultured in Luria–Bertani (LB) liquid medium (Huankai Microbial Co., Ltd., Guangzhou, China) containing 100.0 μg/mL kanamycin at 37 °C and shaken (150 rpm) until OD_600_ reached ~0.6. Isopropyl β-D-1-thiogalactopyranoside (IPTG; Takara Bio Inc., Beijing, China) was added to a final concentration of 0.5 mM and shaken (150 rpm) for 12 h at 25 °C. The cells were then rinsed with PBS and stored at 4 °C for further use.

### 2.4. Pulling down Potential Proteinaceous Ligand Candidates by BSDS

*E. coli* BL21(DE3) with pET28a-inaQn-TB-I.1p (I.1P), pET28a-inaQn-TB-II.4p (II.4P), and pET28a-inaQn-TB (T) were resuspended in 50 mL PBS with 200 μL of the above extracted oyster protein, respectively, incubated at 37 °C, and shaken gently at 80 rpm for 30 min. Cells were collected by centrifugation at 3000× *g* for 5 min, washed twice with PBS, and resuspended in 5.0 mL of digestion buffer (20 mM Tris-HCl and 150 mM NaCl, pH 8.0). The thrombin (Yeason, Shanghai, China) was added at an effective cleaving mass ratio of 1:2000 (2.0 U enzyme per 1.0 mg target protein), incubated for 3 h at 37 °C, and shaken at 120 rpm to release the surface-displayed P protein of GI.1 HuNoV. The supernatant was collected by centrifugation at 12,000× *g* for 2 min at 4 °C and stored at −80 °C for further use.

### 2.5. Identification of Proteinaceous Ligand Candidates

The thrombin-released supernatant of P and T was dissolved in 5 × SDS-PAGE loading buffer (Servicebio, Wuhan, China) and separated in 12% SDS-PAGE gel. Protein bands were collected, cut into small pieces, and rinsed twice with ultrapure water. Fifty percent acetonitrile (ACN) containing NH_4_HCO_3_ (25 mM) was added and shaken for 120 min at 37 °C to decolorize. The ACN was discarded, and 10 mM dithiothreitol (diluted in 25 mM NH_4_HCO_3_) was added. The mixture was incubated at 60 °C for 20 min. A total of 25 mM iodoacetamide (diluted in 25 mM NH_4_HCO_3_) was added and incubated in the dark for 20 min. After rinsing with ACN again, the trypsin (12.5 ng/μL) was added and incubated at 4 °C for 30 min. Then the gel pieces were incubated at 37 °C overnight for enzymatic digestion. After adding 1 mL of extraction solution (60% acetonitrile, 35% deionized water, and 5% formic acid), the mixture was crushed by ultrasound for 5 min and placed at 37 °C for 30 min. The extract was lyophilized at 55 °C for 3 h with a vacuum centrifuge concentrator (Thermo Fisher Scientific, Waltham, MA, USA) and desalted by a micro-desalting column. The desalted sample was dissolved in 10 µL of 0.1% formic acid water, and after 5–10 min the sample was transferred to the sample bottle for further MS analysis.

The protein samples were analyzed by a nanoliter liquid chromatography tandem with a quadrupole orbital trap mass spectrometer (Easy nLC 1200/Q Exactive plus, Thermo Fisher Scientific, Waltham, MA, USA). The sample was dissolved in mobile phase A (0.1% formic acid and 2% acetonitrile) and separated by gradient elution using a mixture of mobile phase A and phase B (0.1% formic acid and 98% acetonitrile) in a liquid chromatography system. The sample (5 μL) was loaded onto the capture column and subsequently separated on the analytical column with a 30 min linear gradient, from 94% mobile phase A to 20% mobile phase A, with a flow of 350 nL/min. The peptide parent ions and their corresponding secondary fragments were detected and analyzed with a high-resolution Orbitrap at an ion source voltage of 1.8 kV. The scan range and resolution of the primary mass spectrometry were 350–1800 *m*/*z* and 70 kHz, respectively. The scan range and resolution of the second mass spectrometry were 200–2000 *m*/*z* and 17.5 kHz. The dynamic exclusion time was set to 30 s to reduce the repeated scanning of the parent ion. Based on the results of the relative quantification, proteins with an abundance ratio of P/T > 5 are considered to be captured explicitly by BSDS.

The mass spectra data were retrieved in the UniProt database using Proteome Discoverer 1.3. Identified proteins in group I.1P but not in group II.4P (positive control) and group T (negative control) were considered as ligand candidates for binding GI.1 HuNoV in oyster tissues, which were further aligned in the UniProt’s *Crassostrea gigas* database. The subcellular localization of proteins obtained by mass spectrometry was predicted by pLoc-mEuk [28].

### 2.6. Prokaryotic Expression and Purification of oTNF and oIFT

The protein signal peptides and transmembrane domains of oTNF and oIFT were predicted using the online tool: SignalP-5.0 and TMHMM-2.0. The sequence analysis results of oTNF and oIFT are shown in Figure A1 and Figure A2, respectively. The amino acids 85–318 of oTNF were predicted to be in the extracellular region. Therefore the first 84 amino acids were considered to be cut off. The theoretical molecular weight of the truncated oTNF was 27.7 kDa. The transmembrane domain prediction results of oIFT showed that the amino acids 210–733 of oIFT were predicted to be in the extracellular region. Therefore the first 209 amino acids were considered to be cut off. The theoretical molecular weight of oIFT after truncation was 62.9 kDa.

According to the analysis results, the nucleic acid sequence of oINF and oIFT was synthesized by Sunny Biotechnology Co., Ltd., Shanghai, China. The sequence was inserted into plasmid pET-28a (+) and transferred to competent *E. coli* BL21(DE3) for expression. The stress-induced expression of the recombinant *E. coli* is the same as the above description. Overnight cultures were centrifuged (7500× *g*, 15 min) to collect the precipitate resuspended in a lysis buffer. The recombinant proteins were named roTNF and roIFT, respectively. Two proteins were purified using Ni Nitilotriacetic acid (NTA) beads 6FF (Smart-lifesciences, Changzhou, China) according to the protocol [19]. The P protein of HuNoV GI.1 and GII.4 was obtained similarly [19].

### 2.7. Evaluation of the Binding Ability of roIFT and roTNF

ELISA was used to evaluate the binding ability of roTNF and roIFT to P proteins of GI.1 and GII.4 HuNoV. The roIFT (62.9 kDa) was diluted to 10.0 μg/mL with a coating buffer (Sangon Biotech Co., Ltd.). The roTNF (27.7 kDa) was diluted to 4.4 μg/mL to ensure the same molar concentration as roIFT. The 2 proteins were coated onto wells (100.0 μL in each well) of ELISA plates overnight at 4 °C. The 1.0 mg/mL type III porcine gastric mucin (PGM, containing HBGAs, Thermo Fisher Scientific, Waltham, MA, USA) and 1.0% BSA were used as the positive and negative controls, respectively. After washing with 150 µL PBS 4 times, 120 µL 1.0% BSA was added to each well and blocked for 1 h at 37 °C. After another 4 washes with PBS, 100 µL P protein (GI.1 or GII.4, 10.0 µg/mL) was added into each well and incubated at 37 °C for 1h. The anti-P protein primary antibodies were obtained in our previous study [22]. All the primary antibodies were incubated with the *E. coli* BL21 (with plasmid pET-28a) lysate for 4 h at room temperature, to rule out its nonspecific recognition of bacterial proteins before use. After washing 5 times with 150 µL TBST (0.1% Tween-20, Sangon Biotech Co., Ltd.), primary antibodies (diluted with 1.0% BSA at a ratio of 1:4000) against the corresponding genotype of the P protein stored in our laboratory were added to each well [22]. After incubation at 37 °C for 1 h, wells were washed with TBST 5 times. The secondary antibody (diluted with 1.0% BSA at a ratio of 1:8000) HRP-conjugated Goat Anti-Rabbit IgG (Sangon Biotech Co., Ltd.) was added to each well and then incubated for 1 h at 37 °C. After 5 washes of TBST, 3,3′,5,5′-tetramethylbenzidine (TMB, Frdbio, Wuhan, China) was added, followed by 10 min of incubation in the dark. The reaction was terminated using 2.0 mol/L H_2_SO_4_ and read with a microplate reader (Thermo Fisher Scientific Co., Ltd.) at 450 nm wavelength. Samples were considered positive when the ratio of OD_450_ of the positive sample to that of the negative control sample (P/N) was equal to or higher than 2.0 [29].

### 2.8. mRNA Level of oTNF and oIFT in Oyster Tissues

RT-qPCR was used to detect the mRNA of oTNF and oIFT in different oyster tissues monthly from December 2020 to September 2021. Primer Premier 6 was used to design qPCR primers according to the nucleic acid sequence of oTNF and oIFT, as shown in Table A1. Frozen oyster tissues were ground with a pre-cooled tissue grinder with 1 mL Total RNA Extractor (Sangon Biotech Co., Ltd., Shanghai, China). RNA was extracted according to the protocol of the Total RNA Extractor. Nanodrop (Thermo Fisher Scientific, Waltham, MA, USA) was used to measure the concentration of extracted RNA. The DNA in the samples was removed by adding Rnase-free ddH2O (11.0 µL) and 4 × gDNA wiper Mix (4.0 µL; Vazyme, Nanjing, China) and keeping it at 42 °C for 2 min. A total of 4.0 µL of 5 × HiScript III qRT SuperMix (Vazyme, Nanjing, China) was added to the reaction solution and incubated at 37 °C for 15 min, followed by reaction at 85 °C for 5 s to achieve the cDNA. The cDNA was diluted 20-fold as a template, and fluorescent quantitative PCR (qPCR) was performed with Universal SYBR qPCR Master Mix (Vazyme, Nanjing, China) according to the protocol shown in Table A2 and Table A3.

### 2.9. Statistical Analysis

Each experiment was performed in triplicate (*n* = 3) and independently repeated more than three times (*n* > 3). Statistics were analyzed by GraphPad Prism version 9 (GraphPad Software, Boston, USA). Differences in means were considered significant when *p* < 0.033.

## 3. Results and Discussion

### 3.1. Identification of the Captured Proteins

After excluding the captured proteins shared with the negative control group (T group), 734 and 485 captured proteins were identified from the I.1P and II.4P groups, respectively. An amount of 356 proteins was present in both groups, and 378 were found only in the I.1P group. The above 378 proteins were further analyzed to discover the specific binding ligands of GI.1 HuNoV. Among them, 148, 179, 184, and 204 proteins were located in the mantle, heart, gills, and digestive glands, respectively (Figure 2). In each tissue, 45, 44, 43, and 50 proteins were unique. A total of 41 captured proteins were found in all tissues. The enormous amounts of captured proteins (204) were located in digestive glands, where the HuNoVs are commonly bioaccumulated [12,13,17].

### 3.2. Selection of Potential Proteinaceous Ligands

In order to narrow down the candidates, we performed a prediction of the subcellular localization of the 204 proteins mentioned above. As pLoc-mEuk would have predicted some of the proteins annotated as cell membrane proteins in UniProt as exocytotic proteins, we retained all of them as the membrane or exocytotic proteins and ended up with 55 candidate proteinaceous ligands (Table 1).

In addition, a protein-protein interaction (PPI) Network analysis of the above 204 proteins was conducted through the STRING database (https://string-db.org/ (accessed on 13 December 2022)) [30]. Due to the limitation of the database, only 167 proteins were included in the network. Among the 167 nodes, 119 nodes were independent. The interactions between the remaining 48 nodes were analyzed and visualized by Cytoscape (Figure 3). The size of the nodes corresponds to the connectivity degree, and the width of the edge represents the betweenness centrality (BC). The connectivity degree is defined as the number of adjacent links, i.e., the number of interactions that connect one protein to its neighbors. BC is the fraction of the number of shortest paths that pass through each node, which measures how often nodes occur on the shortest paths between other nodes. In addition, the predicted membrane or secreted proteins were labeled in red. This network contains nine independent groups, each of which may be captured as a complex by BSDS. In order to screen for potential ligands that interact directly with the P protein of GI.1 norovirus, we did not search for candidate ligands from the group with a high number of protein associations. For example, the Guanine nucleotide-binding protein subunit beta-5 (GNbeta5) with the UniProt number K1RF07 is one of the core targets in the networks. The human GNbeta5 can interact with the hepatitis B virus [31]. However this protein may be pulled down indirectly by the other interacted proteins in the same group.

The membrane protein oIFT (K1QDK8) was selected from an independent group that has only two proteins, as shown in Figure 3. IFT is an essential component of the bidirectional transport system within the cilia, mediating the transport of ciliary proteins from the cytoplasm to the tip of the cilia [32]. IFT is divided into two subunits related to forward and reverse transport, respectively. The IFT has the TRAM–Lag1p–CLN8 (TLC) domain, which has four possible functions: catalyzing the synthesis of ceramide-like moieties or activating lipid synthesis, protecting proteins from proteolysis, lipid transporting, and acting as lipid sensors [33]. However no report about the interaction between the oIFT and foodborne viruses was available.

In addition, oTNF was selected from the other proteins not included in the PPI network. TNF is ubiquitous in all animals and is a member of the tumor necrosis factor superfamily, a class of cytokines with a conserved TNF homology domain at the C-terminus and a type II transmembrane protein [34]. TNF is a pleiotropic cytokine that mediates a wide range of cellular responses and plays a vital role in the subject’s defense against bacterial, viral, and parasitic infections [35]. In vertebrates, TNF can regulate several critical cellular immune processes, such as phagocytosis, apoptosis, cell differentiation, and proliferation [36,37]. Sun et al. discovered a new TNF from Pacific oysters and found that it could regulate apoptosis and phagocytosis in oyster blood cells, activate immune-related enzymes such as phenol oxidase and lysozyme, and regulate antibacterial activity [38]. *Crassostrea hongkongensis* had a constitutive expression pattern of TNF, and the transcript levels of TNF were significantly upregulated after artificial contamination of the oyster with *Vibrio alginolyticus* and *Staphylococcus haemolyticus* [39]. The tandem MS spectra of unique peptides of oTNF (K1QDK8) and oIFT (K1QV98) are shown in Figure 4.

### 3.3. Binding Ability of roTNF and roIFT to P Proteins

The binding ability of roTNF and roIFT to P proteins of GI.1 and GII.4 HuNoV is shown in Figure 5. The OD_450_ of negative control (1.0% BSA) was all less than 0.1. PGM was used as a positive control. PGM (20.0 µg/mL) had positive binding abilities to the GI.1 P protein and GII.4 P protein with a P/N of 2.460 and 3.067, respectively. A higher concentration of PGM (1.0 mg/mL) had stronger binding abilities with a P/N of 2.978 and 4.813, respectively. The roTNF and roIFT also showed positive binding ability with the GI.1 and GII.4 P proteins. The binding ability of roTNF with the GI.1 P protein (P/N 5.885) was slightly higher than the GII.4 P protein (P/N 5.327). The binding ability of roIFT with the GI.1 P protein (P/N 6.974) was significantly higher than that of the GII.4 P protein (P/N 3.949). These results may explain why GI.1 HuNoV was more often detected in oyster tissues [12,20,24]. In addition, with the same molar concentration, the roIFT shows higher binding ability with the GI.1 P protein than the roTNF. However, as the ELISA is a semi-quantitative method, a more accurate method, such as the Biacore SPR system, should be used to compare the binding ability between the above proteins and HuNoV in the follow-up study.

### 3.4. Distribution of oTNF and oIFT in Oyster Tissues

As no commercial antibody against oTNF and oIFT was found, the colocalization of these two proteins and HuNoV particles cannot easily be conducted by the multiplex immunoassays. Therefore, the mRNA level of two proteins in four tissues was detected monthly by RT-qPCR, to evaluate the distribution of these two proteins (Figure 6a,b). Figure 6c,d show the average fold change (mRNA level in tissue/mRNA level in the heart) in four different tissues. As shown in Figure 6c, the mRNA level of oTNF in digestive glands is the highest and is significantly higher than that of the heart (*p* < 0.001). This result is consistent with other studies [40,41]. There was no significant difference in the mRNA level of oTNF between the digestive glands and gills (*p* > 0.033). The mRNA level of oIFT in the digestive glands (Figure 6d) was significantly higher than in other tissues (*p* < 0.001), whereas mantle and gills had no significant difference in mRNA level (*p* > 0.033). As the distribution of the GI.1 HuNoV [26] shows a similar pattern to that of oIFT and oTNF, the oTNF and oIFT may play an important role in the bioaccumulation of GI.1 HuNoV in oysters’ digestive glands.

So far, our lab has identified three potential proteinaceous ligands of HuNoV from oyster tissues: oHSP 70 [19], oTNF, and oIFT. However the contribution of each ligand to the binding of HuNoV in the natural environment is still inconclusive. In addition, the other mined proteins in the study need to be evaluated further. The fluorescence colocalization of a broader range of genotypes of HuNoV and binding ligands needs to be further verified to understand the mechanism of HuNoV accumulation in oysters. Those theoretical bases can help develop new methods to block oyster enrichment or improve oyster purification in the future.

## 4. Conclusions

In this study, the BSDS of GI.1 and GII.4 noroviruses were used to identify potential proteinaceous ligands in four tissues of the oyster: gill, mantle, digestive gland, and heart, and a library of 378 proteins was initially obtained by mass spectrometry. Subsequent screening yielded 55 proteinaceous ligands candidates located in the digestive gland. We finally selected two proteins, oTNF and oIFT, for subsequent validation and demonstrated that the recombinant proteins of both had higher binding abilities for G I.1 than G II.4 HuNoV, even though all of them are positive. We then measured the mRNA levels of both proteins in different tissues and showed that their expression was highest in digestive glands and lowest in heart tissues, which is also consistent with the enrichment pattern in oyster tissues for GI.1 HuNoV. Therefore we tentatively concluded that oTNF and oIFT are potential proteinaceous binding ligands in Pacific oyster tissues.

## Figures and Tables

**Figure 1 viruses-15-00631-f001:**
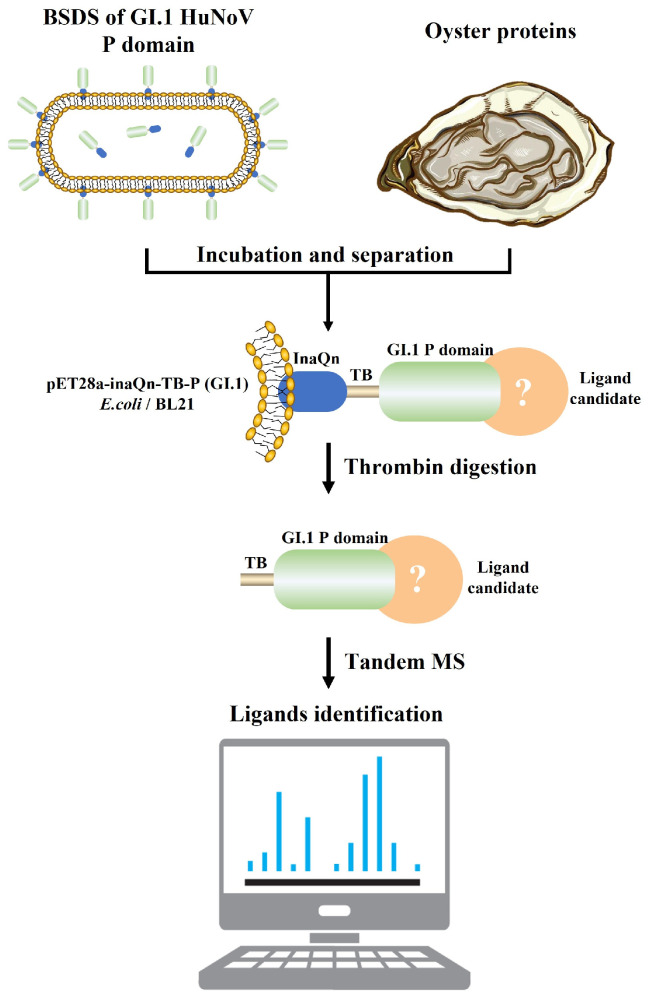
Brief workflow for the separation and identification of GI.1 HuNoV proteinaceous ligands in oyster tissues by BSDS. The BSDS was constructed and incubated with extracted oyster protein to pull down the potential proteinaceous ligand. The complexes of the P domain and ligand candidate were then released by thrombin. The complexes were then identified by a nanoliter liquid chromatography tandem with a quadrupole orbital trap mass spectrometer.

**Figure 2 viruses-15-00631-f002:**
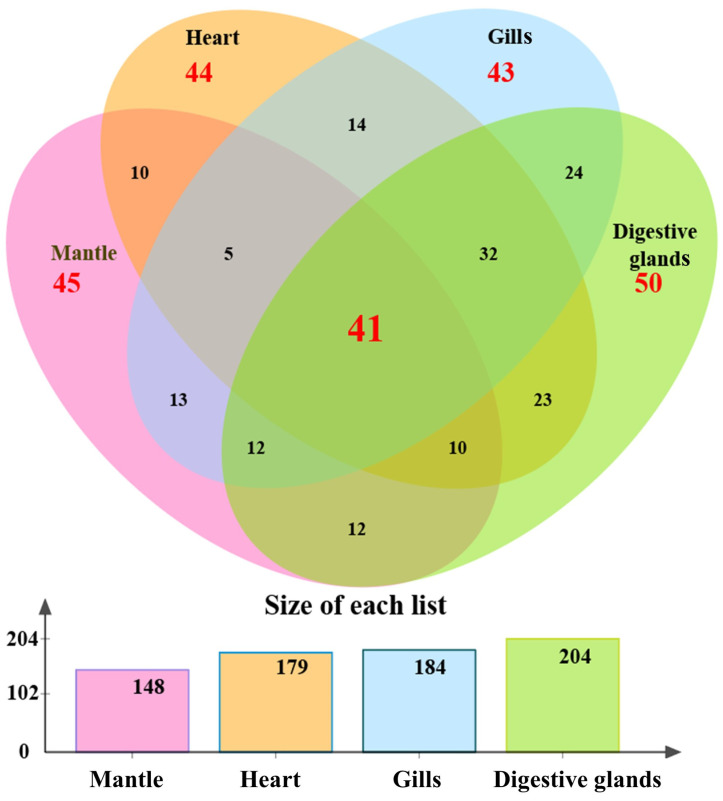
The distribution of 378 differential proteins in four tissues obtained by BSDS I.1P. Among them, 148, 179, 184, and 204 proteins were located in the mantle, heart, gills, and digestive glands, respectively. The amounts of 45, 44, 43, and 50 proteins were unique in each tissue. A total of 41 captured proteins were found in all tissues.

**Figure 3 viruses-15-00631-f003:**
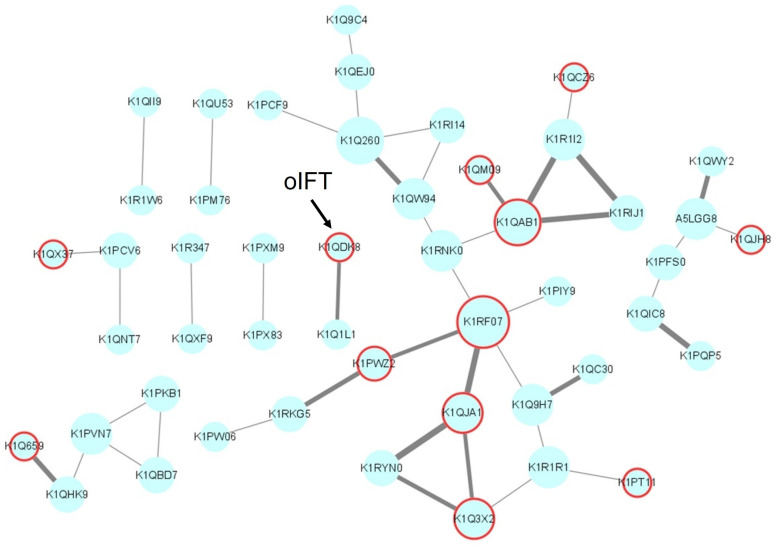
The PPI network of 48 potential viral ligands visualized by Cytoscape. Each node represents the protein’s UniProt login number. The size of the nodes corresponds to the connectivity degree, and the width of the edge represents the betweenness centrality. The membrane or secreted proteins are labeled in red. This network contains nine independent groups, each of which may be captured as a complex by BSDS.

**Figure 4 viruses-15-00631-f004:**
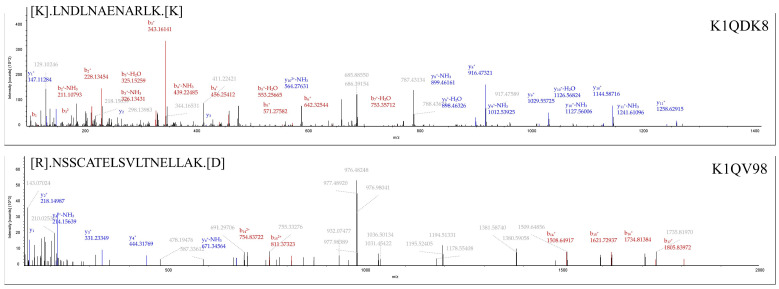
Tandem MS spectra of unique peptides of oTNF (K1QDK8) and oIFT (K1QV98), respectively. The identified peptides are marked on the top left of each MS spectra. The UniProt accession number of the corresponding protein is shown in the top right corner.

**Figure 5 viruses-15-00631-f005:**
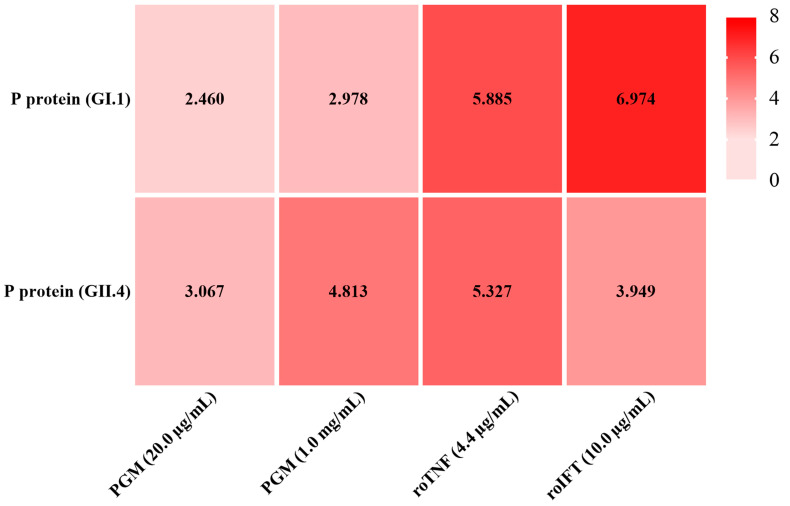
The binding ability of PGM, roTNF, and roIFT to GI.1 and GII.4 HuNoV P proteins, respectively. The OD450 of 1.0% BSA, negative control, was all less than 0.1. PGM was used as a positive control. Taking the criterion of positive to negative (P/N) of ≥2.0.

**Figure 6 viruses-15-00631-f006:**
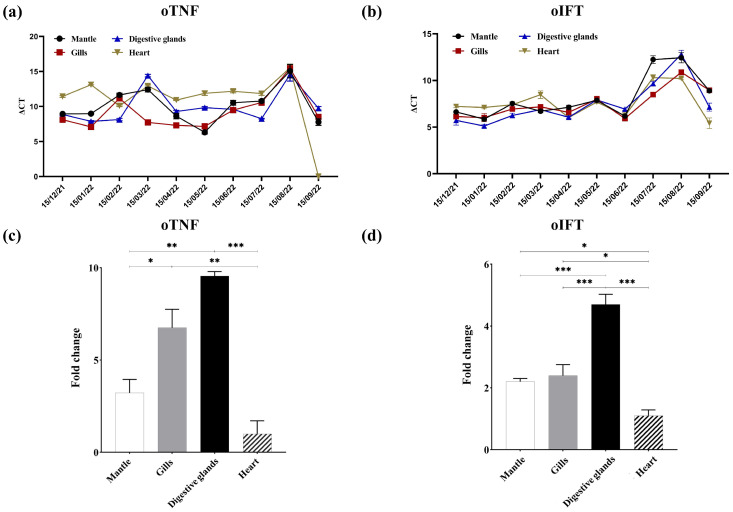
The mRNA levels of oTNF (**a**) and oIFT (**b**) in four oyster tissues from December 2021 to September 2022. The average fold change (mRNA level in tissue/mRNA level in the heart) of mRNA levels of oTNF (**c**) and oIFT (**d**) in four different tissues. (*, *p* < 0.033; **, *p* < 0.002; ***, *p* < 0.001).

**Table 1 viruses-15-00631-t001:** Details of the 55 potential proteinaceous ligands of GⅠ.1 HuNoV.

Uniprot Number	Protein Name	Coverage (%)	Number of Specific Peptides	Molecular Weight (kDa)	Theoretical Isoelectric Point	Tissue Distribution Location
K1P4E2	Putative tyrosinase-like protein tyr-3	4	1	66.7	9.22	MDGH
K1QJ28	Mammalian ependymin-related protein 1	3	1	40.5	6.07	MDGH
K1QN64	Transporter	2	1	95.6	7.81	MDGH
K1PT11	Collagen alpha-2(I) chain	2	1	168.9	5.24	MDGH
**K1QV98**	**TNF_2 domain-containing protein**	6	1	35.8	9.17	MDGH
K1QHF1	Collectin-12	6	1	43.9	8.44	MDGH
K1R7M0	SMC_N domain-containing protein	2	1	88.1	5.26	D
K1RIE6	Small conductance calcium-activated potassium channel protein 2	7	1	43.8	9.11	D
K1PNV2	Trithorax group protein osa	4	1	72.5	9.54	D
K1PRP1	TPR_REGION domain-containing protein	1	1	182.6	5.68	D
K1QHW7	Microfibrillar-associated protein 1	3	1	67.3	5.03	D
K1QHP5	Carbohydrate sulfotransferase 15	2	1	168.4	8.28	D
K1QX85	Bestrophin homolog	3	1	82.3	6.39	D
K1QIT7	Extracellular matrix protein FRAS1	1	1	233.6	5.27	D
K1RF07	Guanine nucleotide-binding protein subunit beta-5	6	1	40.5	6.16	D
K1QX37	phosphopyruvate hydratase	1	1	127.3	7.34	D
K1P919	Protocadherin-like wing polarity protein stan	7	1	39.5	4.7	D
K1PJY2	Inositol polyphosphate 1-phosphatase	3	1	82.3	5.33	D
K1QWW4	Lin-54-like protein	3	1	81.6	8.43	D
K1PUP1	N-acetylated-alpha-linked acidic dipeptidase 2	1	1	84.8	6.34	D
K1PB63	DUF19 domain-containing protein	5	1	19	6.51	D
F8RP10	Bactericidal permeability increasing protein	6	1	52.8	9.76	D
K1QM09	Contactin	3	1	88.7	6.84	D
K1R0F5	Prominin-1-A	3	1	91.2	4.94	D
K1RGD2	Pancreatic lipase-related protein 1	3	1	56.1	6.62	D
K1QH82	Transient receptor potential cation channel subfamily M member 8	2	1	150	6.54	D
K1QM30	EGF-like domain-containing protein	8	1	39.7	7.24	DGH
K1QCZ6	Proprotein convertase subtilisin/kexin type 4	1	1	62.2	7.87	DGH
K1Q3 × 2	Plexin-A4	7	1	64.9	6.24	DGH
K1R1E8	Fibronectin type-III domain-containing protein	22	1	17.1	8.16	DGH
K1QJQ0	Dynein regulatory complex protein 10	4	1	60.7	9.26	DGH
K1QQ05	Insulin-like growth factor-binding protein complex acid labile chain	2	1	104.1	8.79	DGH
K1PKY4	Sodium/calcium exchanger 3	3	1	64.5	5.36	DGH
K1QUL1	Zinc finger CW-type PWWP domain protein 1-like protein	2	1	102.7	5.43	DGH
K1P915	Sushi domain-containing protein	7	1	32.5	5.49	DGH
K1PAY0	Sodium bicarbonate transporter-like protein 11	6	2	98.8	6.71	DGH
K1S3D5	Solute carrier family 22 member 16	20	1	17.6	7.91	DGH
K1QYC3	P-type Cu(+) transporter	4	2	131.3	6.3	DGH
K1Q7G4	Protein LAP2	3	1	80	4.64	DG
K1PRW3	Innexin	5	1	52.4	8.53	DG
K1QJH8	Amiloride-sensitive cation channel 2, neuronal	2	1	72.2	6.61	DG
K1QJA1	Cell division control protein 42-like protein	10	1	22.7	5.62	DG
K1R5R3	DBH-like monooxygenase protein 2-like protein	2	1	126.4	6.02	DH
K1QGG3	Secreted protein	22	1	16.6	5.97	DH
K1QUW2	Pancreatic trypsin inhibitor	3	1	134.3	10.61	DM
K1RYS4	E3 ubiquitin-protein ligase TRIP12	1	1	109.4	9.54	DM
K1RRI7	Ficolin-2	10	1	42.5	6.37	DM
K1Q659	Centromere protein F	1	2	443.5	4.94	DGM
K1QUK9	Migration and invasion-inhibitory protein	2	1	74.7	8.16	DGM
K1PWZ2	Metabotropic glutamate receptor 8	3	1	79.8	7.52	DMH
K1QAB1	AP-2 complex subunit alpha	2	1	109.6	7.78	DMH
**K1QDK8**	**Intraflagellar transport protein 74-like protein**	2	1	86.1	5.43	DMH
K1PBI1	Metalloendopeptidase	2	1	91.6	6.52	DMH
K1RVV6	Titin-like	2	1	76.9	8.82	DMH
K1QIA5	Membrane progestin receptor beta	2	1	103.3	7.83	DMH

M: mantle; G: gills; D: digestive glands; H: heart.

## Data Availability

Please contact the authors for raw experimental data.

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
