# Peer review of "Identification of Potential Proteinaceous Ligands of GI.1 Norovirus in Pacific Oyster Tissues"

_viruses, 2023, doi:10.3390/v15030631_

Round 1

Reviewer 1 Report

This report by Lyu et al. describes work to establish the identity of proteinaceous ligands in oyster tissue that interact with the P domain of the capsid proteins of GI.1 norovirus.  The experimental approach employed a bacterial cell surface display system (BSDS), previously used by this team to identify binding ligands to norovirus such as the oyster heat shock protein 70.  Two new binding ligands were identified in oysters, oTNF and oIFT.  

A few comments seeking to improve the manuscript are as follows:

1.     The authors selected binding ligands oTNF and oIFT for further investigation after stating that they researched the literature (line 246).  Table 1 lists the UniProt ID number of the other potential ligands.  Would it be possible to insert a column that briefly IDs the protein?  For example, K1P4E2 is tyrosinase-like protein and K1QJ28 is Mammalian ependymin-related protein 1. If this annotation is not feasible, it would be helpful to highlight the oIFT (K1QDK8) and oTNF (K1QV98) numbers on the table so that the reader can easily see the tissue distribution and other parameters given in the table. 

2.     Although an excellent expression system, it is sometimes difficult to remove all bacterial impurities from recombinant proteins expressed in bacteria.  Were the antibodies raised against the norovirus P proteins used in the ELISA generated against P domains expressed in bacteria?  Please clarify the controls to rule out nonspecific recognition of bacterial proteins in the ELISA associated with the purification of the oTNF and oIFT ligands in bacteria.  In addition, how was it ruled out that the P domain antigen itself was not binding to the ELISA plate directly?  

3.     The discovery would be enhanced if the investigators could provide additional evidence of a direct interaction between the norovirus capsid and the new binding ligands.  Have other strategies been tried, such as pull-down assays or yeast two-hybrid systems?  This would also allow mutagenesis to map the precise binding interactions. 

4.     The authors mention that the role of these newly identified binding ligands could not be verified experimentally at this point.  But it would be interesting to have some additional discussion on the rationale for exploring these proteins above the others on the table, and why these particular proteins might be targets for binding in the oysters. 

Minor:

Line 289.  “than that of the heart”

Reviewer 2 Report

The authors report on the identification of potential proteinaceous ligands for norovirus GI.1 in oyster tissues. They followed an approach that previously led to the identification of an HBGA-like ligand in romaine lettuce and heat shock protein in oysters. The P domain of norovirus GI.1 was displayed on the surface of BL21 E. coli, linked by a thrombin cleavable tag. After incubation with different oyster tissues the P protein was cleaved off and processed for MS analysis to identify any bound ligands. The study focused on two potential protein ligands, oTNF and oIFT, from a large set of candidates. The authors demonstrated binding of norovirus P domain to recombinant versions of these targets and evaluated the distribution of these targets in oyster tissues. 

Major comments:

Suggested change in title: 

Identification of potential proteinaceous ligands of GI.1 norovirus in Pacific oyster tissues

In the concluding paragraph the authors refer to potential ligands, therefore the amended title would better represent the data.

Binding assays: Could the authors explain why they did not perform a binding experiment with a dilution range of the recombinant oyster proteins to generate binding curves? Such an experiment would certainly enhance the paper and illustrate the difference between GI.1 and GII.4 binding well and should be included. 

Minor comment:

The manuscript needs to be checked carefully for English errors and would benefit from editing by a native English speaker. 

Reviewer 3 Report

The research article by Lyu et al. used a bacterial cell surface display system to identufy 55 ligands that can bind to GI.1 human norovirus. They then specifically showed binding of GI.1 P protein with two oyster proteins, oTNF and oIFT. The binding may also explain HuNov bioaccumulation in oysters. The article is well organized with sound, step-wise experiments. There are multiple grammatical errors throughout which can be corrected. Here are some possible corrections.

Figure 1. Please add a brief description of the workflow.

Figure 2. Please describe the caption.

Table 1. Adding protein names in the table will be very informative.

Figure 4. Why were the specific concentrations for roTNF and roIFT chosen? Testing binding affinity over a range of concentrations would have been more informative.

Figure A1 and A2. Please briefly describe the captions.

Line 106: Italicize “Pseudomonas syringae”.

Line 195: “washed” should be replaced with “washing”.

Line 208: Kindly explain the data analysis performed, as “when positive to negative” phrase is unclear.

Line 229: “deletion” can be replaced with “exclusion”.

Line 245: After the mention of ‘table 1’ in the text, certain proteins from table 1 need to be discussed for their significance/potential/use as ligands in previous literature.

Line 246: “After researching the literature” – these details regarding the mentioned proteins from the literature can be described in the Discussion section. Additionally, better justification for short listing these two proteins (oTNF and oIFT) need to be provided.

Additional comments:

Please add a statistical analysis subtopic in the methods and name the statistical tests you used for different experiments there.

A Discussion section will be beneficial.
